# Subchronic Toxicity of GmDREB3 Gene Modified Wheat in the Third Generation Wistar Rats

**DOI:** 10.3390/plants11141823

**Published:** 2022-07-12

**Authors:** Jie Tian, Xiang-Hong Ke, Yuan Yuan, Wen-Xiang Yang, Xiao-Qiao Tang, Lan-Jie Pei, Jun Fan, Qin Zhuo, Xiao-Guang Yang, Jia-Fa Liu, Bo-Lin Fan

**Affiliations:** 1Hubei Provincial Key Laboratory for Applied Toxicology, Hubei Provincial Center for Disease Control and Prevention, Wuhan 430079, China; jiessie_tian@163.com (J.T.); cilan-724@163.com (X.-H.K.); emily_fighting@163.com (Y.Y.); ywx_21@163.com (W.-X.Y.); melon_qiao@163.com (X.-Q.T.); peilanjie123@163.com (L.-J.P.); u571_f16@163.com (J.F.); jfllionm@sohu.com (J.-F.L.); 2Key Laboratory of Trace Element Nutrition of National Health Commission, Chinese Center for Disease Control and Prevention, Beijing 100050, China; zhuoqin@ninh.chinacdc.cn (Q.Z.); xgyangcdc@163.com (X.-G.Y.)

**Keywords:** subchronic toxicity, GmDREB3, genetically modified wheat, rats

## Abstract

The aim of the current study was to evaluate the subchronic toxicity of GmDREB3 gene modified wheat in the third generation rats. SPF Wistar rats were fed with transgenic wheat diet (Gm), parental wheat diet (Jimai22) and AIN-93 rodent diet (Control), respectively, for two generations, to produce the third generation rats which were used for this study. The selected fresh weaned offspring rats (20/sex/group) were given the same diet as their parents for 13 weeks. No toxicity-related changes were observed in rats fed with Gm diet in the following respects: clinical signs, body weights, body weight gains, food consumption, food utilization rate, urinalysis, hematology, serum biochemistry and histopathology. The results from the present study demonstrated that 13 weeks consumption of Gm wheat did not cause any adverse effects in the third generation rats when compared with the corresponding Jimai22 wheat.

## 1. Introduction

Wheat, as one of the main food crops (maize, rice, wheat), is widely planted on about 220 million ha area of at least 43 countries in the world [1,2]; the global wheat output is estimated to increase by 1.0% every year [3]. Wheat is rich in starch, fat, protein and microelements and provides total food calories for about 40% of the world’s population [4,5]. However, the output of wheat is seriously influenced by climatic changes and water scarcity in the environment. In the future, with the global temperature increasing, the frequency of drought occurrence will be greater than ever [6]. Jimai 22 wheat is a native crop mainly cultivated in northern China [7], but although it possesses some clear advantages of high yield, strong environment adaption and strong anti-reversion, it fails to be well promoted due to its poor resistance to drought. Consequently, to improve the drought tolerance of wheat is an urgent task in China and also around the world. 

For many years, scientists have experimented with different ways to enhance drought resistance in crops, such as traditional hybridization methods and genetic engineering technology. Conventional hybridization has been quite effective to address the complex quality traits such as adaption to changing climate conditions, which are usually mediated by multiple genetic factors; however, it requires a huge commitment of time and human resources [8,9]. Genetic engineering technology overcomes some disadvantages of traditional breeding and selection methods, and therefore it is also applied as a promising way to develop new cultivars in modern agriculture to transfer beneficial genes from one organism to another to improve a certain trait of crops [10]. In this research, the GmDREB3 gene from soybean was inserted into the wild wheat (Jimai 22) to enhance its tolerance to drought. This Gm wheat was demonstrated to have drought resistance and can be planted in semi-arid and arid areas, but as a new transgenic crop it still faces the challenges connected with assessment in food safety for the reason of exogenous gene insertion. The unexpected toxic ingredients, reduction of the nutritional value and other unintended effects of this Gm wheat are still public concerns. Consequently, toxicity assessment is necessary to ensure safety before Gm Wheat can be accepted as food or feed into market [11,12,13].

Based on the previous two-generation reproduction studies of this Gm wheat [14,15], we performed this 13-week subchronic toxicity study in SPF grade Wistar rats, whose ancestors were fed with Gm wheat for their entire life cycle over two generations, to evaluate the potential adverse effects. This research followed OECD Guideline 408 for repeated dose 90-day toxicity study in rodents [16] and good laboratory practice standards (GLP standard).

## 2. Results

### 2.1. Clinical Observations, Body Weights and Food Consumption

Throughout the experiment, there were no toxicity-related signs observed in any group. The increase trends of body weights in the three groups were similar. Compared with the Jimai22 female group, the body weights of the Gm female rats showed no significant differences except during weeks 2, 3, 4 and 6 (*p* < 0.05 or *p* < 0.01). The body weights exhibited no difference between Gm male rats and Jimai22 male rats. In addition, no significant differences were observed in weekly food consumption of rats in the two wheat groups. The results are shown in Figure 1 and Figure 2. 

There were no significant differences between the Gm group and Jimai22 group (*p* > 0.05) in total body weight gains, total food consumption or total food utilization rate. Compared with the AIN-93 control group, the total body weight gains and total food utilization of the Gm group (females and males) were both significantly higher (*p* < 0.01, *p* < 0.01), but total food intake of Gm females and Gm males displayed no difference (*p* > 0.05). The results are shown in Figure 3, Figure 4 and Figure 5. 

### 2.2. Urine Analysis

There were no statistically significant differences in any parameters of Gm female and Gm male rats compared with the Jimai22 or AIN-93 control groups (data not shown).

### 2.3. Hematology and Serum Biochemistry

Gm males had lower Hb, HCT % and a higher CHOL (*p* < 0.05, *p* < 0.01) and Gm females had a lower BUN and a higher CHOL (*p* < 0.05, *p* < 0.01), when compared with the Jimai22 group. When compared to the control group, Gm females had lower HCT%, BUN, ALT and AST, and higher Glu, CHOL, TP and Ca^2+^; Gm males had lower HCT%, AST, Na^+^ and Cl^−,^ and a higher Glu (*p* < 0.01 or *p* < 0.05); the other serum biochemistry indices presented no significant differences among the three groups. The results are shown in Table 1 and Table 2. 

### 2.4. Organ Weights and Relative Organ Weights

When compared with the Jimai22 group, Gm female rats had higher absolute weights and relative weights of their liver and adrenal glands (*p* < 0.01 or *p* < 0.05). When compared with the control group, Gm female rats had higher absolute weights of heart, liver, spleen, kidneys and thymus (*p* < 0.05 or *p* < 0.01), and lower relative weights of brain and ovaries; Gm male rats had lower relative weights of heart and brain (*p* < 0.01 or *p* < 0.05). The results are shown in Table 3 and Table 4. 

### 2.5. Histopathology

There were some histopathological changes in three group animals, including: slight interstitial inflammation in the lungs; slight stomach glandular dilatation; slight inflammatory cell infiltration and slight vacuolation in livers; slight interstitial inflammatory infiltrate, hyaline casts and calcium deposition in kidneys; testis atrophy and sperm loss in epididymis. Details are shown in Table 5 and Figure 6. Histopathological results in all tissues did not reveal any significant difference between the Gm and Jimai22 groups. 

## 3. Discussion

Dehydration responsive element-binding proteins (DREBs) are members of a larger family of transcription factors (TFs), which are involved in the resistance to abiotic stress-induced oxidative damages by regulating the expression of genes in the stress defense pathway [17,18]. In addition, DREBs participate in the induction of salinity tolerance by acting on the downstream (auxin and ethylene signaling pathways) and upstream (ABA-independent signaling pathway) [19]. DREBs are also involved in immunity against several biotic stresses by modulating the expression of several downstream genes of the defense signaling cascade [20]. Therefore, DREB genes were used widely to genetically engineer plants for resistance to abiotic factors. The Gm wheat used in this study is a new variety produced by genetic engineering technology to introduce the GmDREB3 gene to the original wheat.

Previous nutritional analysis showed that this wheat was nutritionally equivalent to its non-Gm counterpart [21]; preceding studies also showed that this Gm wheat was safe in immunotoxicity studies [15,22], and in two-generation reproduction studies [14,15]. According to the OECD Test Guidelines and the Guidelines for Safety Assessment of Genetically Modified Plants of China, 90-day subchronic toxicity experiments in rodents are generally considered necessary for safety evaluation of all genetically modified foods [23,24,25], but routine 90-day subchronic toxicity assessment in transgenic foods were always conducted in single generation animals. Some studies suggested that long-term and multi-generational assessments are necessary in some cases to further assure the safety of Gm food/feed [26,27]. So far, no 90-day subchronic toxicity study has been performed to address the potential toxicological effects in animals exposed to GM feed over three generations. Consequently, a 13-week subchronic toxicity study was conducted in Wistar rats whose parental animals were pre-exposed to the Gm wheat for two generations.

In the present study, the Jimai22 group was used to test whether this Gm wheat would create any toxic effects due to the insertion of the GmDREB3 gene, while the AIN-93 basic diet was used as a nutritional control to analyze whether the wheat in the diet would affect the nutritional status of the rats. During the period of the 13-week study, compared to Jimai22 group, the body weights of the Gm female group were slightly higher in weeks 2, 3, 4 and 6, which was considered a normal body weight fluctuation. In addition, no remarkable difference was detected in the weekly food consumption, total body weight gains, total food consumption, or total food utilization rate, which also confirmed that the above-mentioned differences were of no toxic concern.

In hematology and serum biochemistry, compared with the Jimai22 group Gm male rats had lower HCT% and Hb and a higher CHOL, Gm females had a lower BUN and a higher CHOL. When compared with the control group: Gm females had lower HCT%, BUN, ALT and AST, higher Glu, CHOL, TP and Ca^2+^; Gm males had lower HCT%, AST, Na^+^, Cl^−^ and a higher Glu. All of the above-mentioned differences were considered to have no toxicological significance since these values were similar to those of the control group and the Jimai22 group; in addition, these observed differences were within the normal reference ranges for rats of this age in our laboratory. Serum total T4, T3 and TSH should be measured on samples obtained from each animal according to OECD 408 (2018 revised version), but these indexes was absent in this study because the investigation was performed in 2016. We acknowledge it was a limitation of the study. We will strictly refer to OECD guideline 408 (2018 version) in the future study of 90-day subchronic toxicity.

With respect to organ weights, significant increases were presented in absolute and relative weights of livers and adrenal glands in female rats of the Gm group, as compared with the Jimai22 group, which were considered to be not treatment-related since the respective findings were not supported by the biochemical or histopathological data generated in our study. Compared with the control group, Gm female rats had higher absolute weights of heart, liver, spleen, kidneys and thymus, and lower relative weights of brain and ovaries; also, Gm male rats had lower relative weights of heart and brain, which were attributed to increased body weights (females and males in Gm group) at the time of necropsy and were considered to have no toxicological significance.

In the histopathological analysis, some kinds of lesions were observed in the Gm group, but the severity and frequency of these lesions were comparable between the Gm and Jimai 22 group. Furthermore, the incidences of these findings were all within the normal reference range for such data established in our laboratory. Thus, all of the above-mentioned histopathological alterations in the Gm group were interpreted as incidental and were considered to be of no significance related to toxicity.

## 4. Materials and Methods

### 4.1. Plant and Diet 

The Gm and Jimai22 wheats were provided by the Chinese Academy of Agricultural Sciences. Both seeds were processed into flour, then the two kind of wheat flour were formulated into rodent diets; both were at a percentage of 69.55% as previous diets of two generations, and meanwhile, the AIN-93 diet was set as a nutritional control [28]. The target genes in Gm, Jimai22 and control diets were detected using the PCR method [14].

### 4.2. Experiment Animals and Breeding Condition

In total, 120 specific pathogen-free (SPF) Wistar rats (60 males and 60 females) were obtained from previous two-generation reproductive experiments. The rats were housed in a good experimental environment and could eat and drink ad libitum. All experimental processes were approved and supervised by the Hubei Provincial Animal Management and Use Committee (NO. 2015022).

### 4.3. Study Design and Administration

The previous two generations of rats (F0 and F1 generation) were fed Gm diet, Jimai22 diet and AIN-93 control diet for their lifetime. A total of 20 males and 20 females of the third generation rats (F2 generation) were randomly selected from each group and were given the same diet as their parents for 13 weeks, to investigate the subchronic toxicity of this Gm wheat. 

### 4.4. Clinical Observations, Body Weight and Food Consumption

All animals were observed daily for signs of toxicity and the availability of food. Animal body weights and food consumption of each group were recorded per week; total body weight gains, total food consumption and total food utilization rate for weeks 1–13 were calculated. 

### 4.5. Urinalysis

Urine samples of all rats were collected by metabolism cages one day prior to necropsy. During the process of urine collection water was available to all rats freely. After observing and recording the urine appearance and volume, the urine indices, including BIL, UBG, KET, PRO, NIT, GLU, LEU, BLD, SG and PH, were determined using an automatic urine analyzer (Dirui; Changchun; China).

### 4.6. Hematology and Serum Biochemistry

At the end of week 13, all animals of the three groups were fasted overnight for 16 h and were given drinking water as before, then the blood samples were collected from abdominal aorta under anesthesia. RBC, WBC, PLT, HCT, Hb, NEUT%, LYMPH%, MONO%, EO% and BASO% were determined using an automatic blood analyzer XT2000iv (Sysmex Corporation; Godo; Japan). PT and APTT were measured using a coagulation function analyzer CA510 (Sysmex Corporation; Godo; Japan). GLU, BUN, CREA, CHOL, TG, ALT, AST, TP, ALB, ALB/GLO, Na^+^, K^+^, Cl^−^ and Ca^2+^ were measured using an automatic analyzer AU680 (Beckman Coulter, Inc.; California; America).

### 4.7. Histopathology

All animals were sacrificed and subsequently submitted to detailed necropsy and histopathology examination. Brain, heart, liver, spleen, kidneys, adrenal glands, thymus, ovaries (female), uterus (female), testes (male), epididymides (male) and prostate (male) were weighed, and the relative organ weights were calculated. The following organs were examined under a microscope: brain, pituitary, spinal cord, sciatic nerve, thyroid, parathyroid, salivary gland, sternum, thymus, oesophagus, trachea, heart, lung, stomach, spleen, liver, duodenum, jejunum, ileum, colon, rectum, pancreas, adrenals, kidneys, aorta, ovaries, uterus, mammary gland, testes, epididymides, prostate, urinary, bladder skeletal muscle, skin, eyes, and lymph node. In the histopathological analysis, the incidence and semi-quantitative score system used was recommended by Shackelford et al. [29]. The degree of lesions in each item was graded from one to five depending on severity: 1 = minimal (<1%); 2 = slight (1–25%); 3 = moderate (26–50%); 4 = moderate/severe (51–75%); 5 = severe/high (76–100%).

### 4.8. Statistical Analysis

Quantitative data in this study including animal body weights, food consumption, organ weight, hematology and biochemistry data were analyzed using one-way analysis of variance (ANOVA), then followed by Student’s *t*-test when the variance of three group data was homogeneous. Urine data were analyzed using the Kruskal–Wallist test, and histopathological changes were analyzed using Chi-square test and Fisher’s exact test. Statistically significant differences were considered if the *p*-value between the groups was less than 0.05. 

## 5. Conclusions

The results of the 13-week subchronic toxicity study of the Gm wheat did not exhibit any toxic effects in Wistar rats, whose ancestors had been pre-exposed to the same wheat for two generations. This suggests that Gm wheat is as safe for food as its parental wheat in our current study.

## Figures and Tables

**Figure 1 plants-11-01823-f001:**
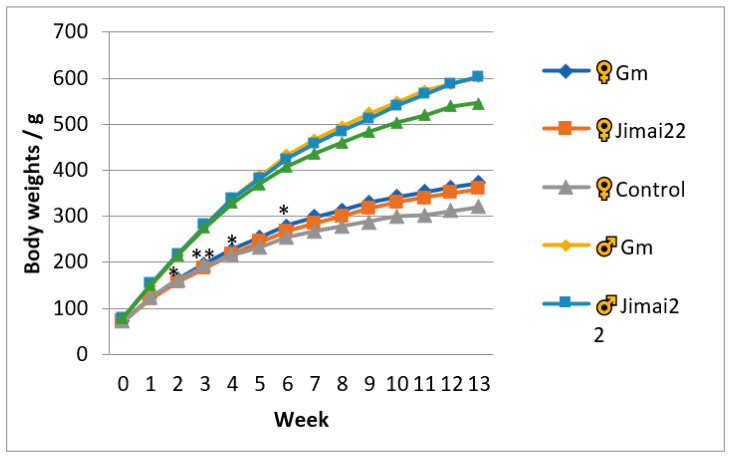
Weekly body weights of female and male animals in three groups. * represents *p* < 0.05 compared with Jimai22 group; ** represents *p* < 0.01 compared with Jimai22 group.

**Figure 2 plants-11-01823-f002:**
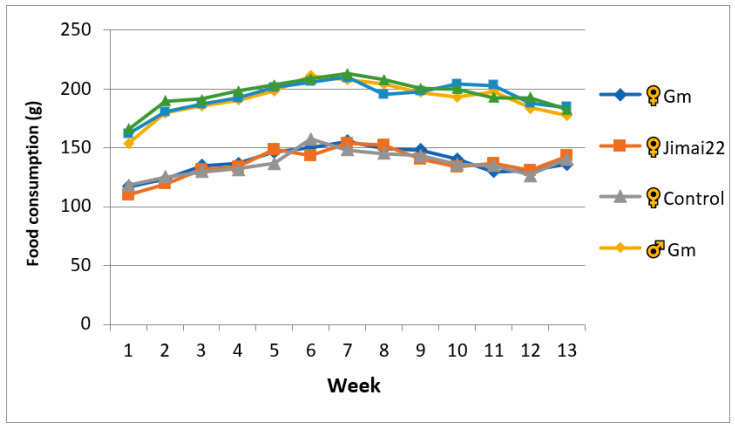
Weekly food consumption of female and male animals in three groups.

**Figure 3 plants-11-01823-f003:**
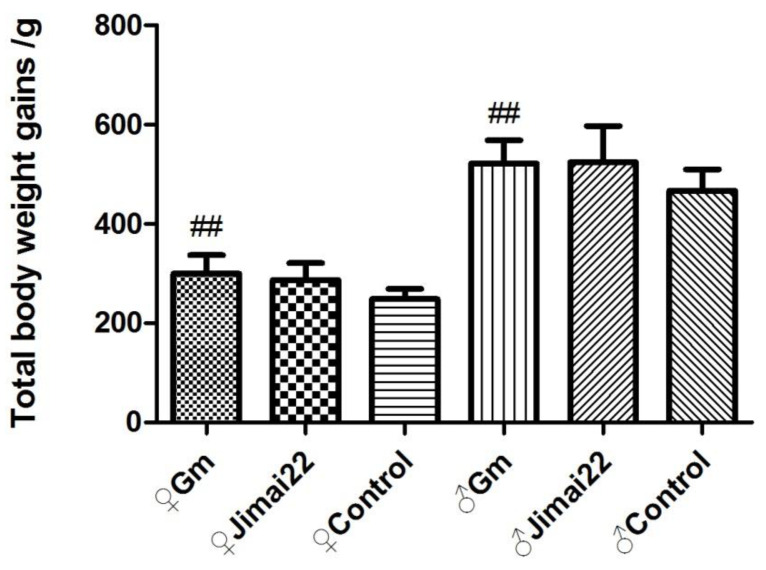
Total body weight gains of females and males during 13 weeks. ^##^ represents *p* < 0.01 compared with control group.

**Figure 4 plants-11-01823-f004:**
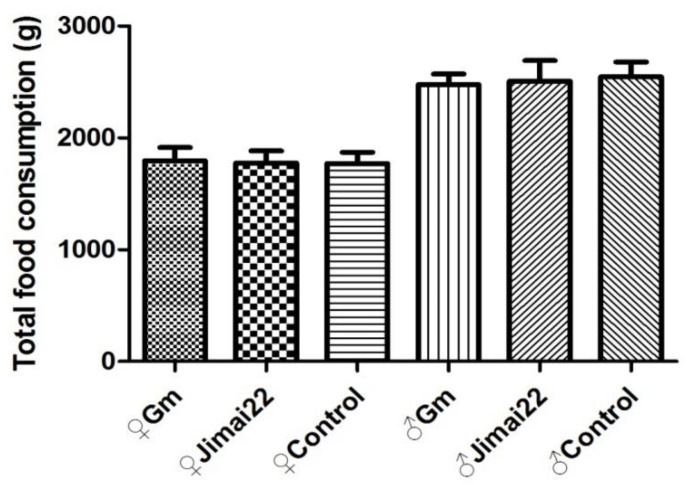
Total food consumption of females and males during 13 weeks.

**Figure 5 plants-11-01823-f005:**
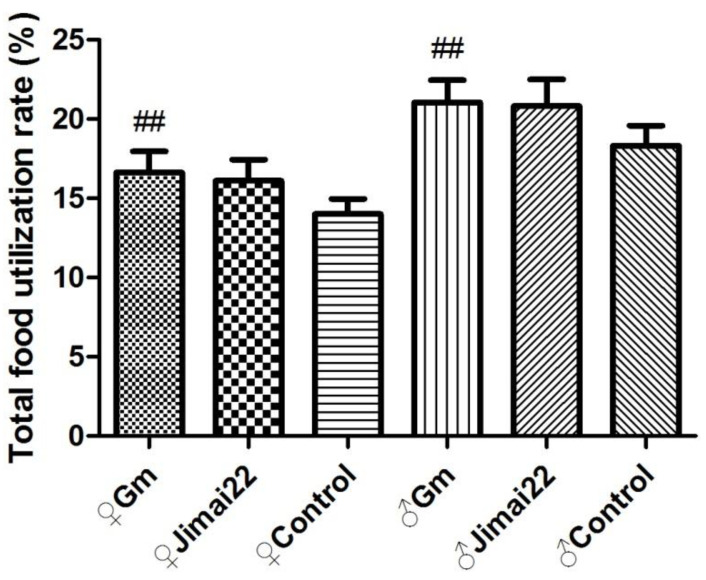
Total food utilization of females and males during 13 weeks. ^##^ represents *p* < 0.01 compared with control group.

**Figure 6 plants-11-01823-f006:**
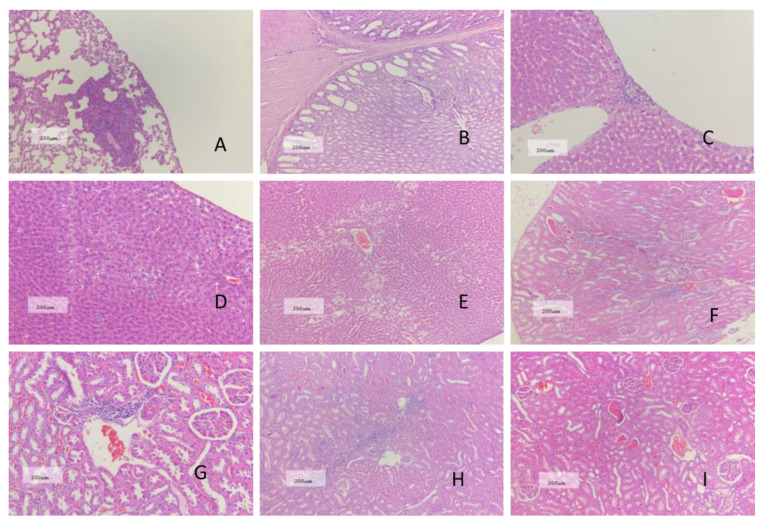
Non-specific histopathological changes in rats at week 13. (**A**): slight interstitial inflammation of lung in Jimai22 group; (**B**): Slight glandular dilatation of stomach in Jimai22 group; (**C**): slight inflammatory cells infiltration of liver in Jimai22 group; (**D**,**E**): slight hepatic vacuolation in Jimai22 and Gm group; (**F**–**H**): slight renal interstitial inflammatory cell infiltration in Control, Jimai22 and Gm group, respectively; (**I**,**J**): Slight renal hyaline casts in Jimai22 group and Gm group; (**K**–**M**): slight renal calcium deposition in Control, Jimai22 and Gm group, respectively; (**N**–**P**): testicular atrophy in Control, Jimai22 and Gm group, respectively; (**Q**,**R**): epididymal sperm loss in Control, Jimai22 group.

**Table 1 plants-11-01823-t001:** Hematological parameters of male and female rats ($\bar{x}$ ± s, *n* = 20).

Sex	Group	WBC	RBC	Hb	LYMP	MONO	NEUT	EO	BAS	HCT	PLT	APTT	PT
		(×10^9^/L)	(×10^12^/L)	(g/L)	(%)	(%)	(%)	(%)	(%)	(%)	(%)	(s)	(s)
	Gm	2.6 ± 1.1	7.66 ± 0.51	134 ± 32	70.0 ± 6.1	2.3 ± 0.9	25.8 ± 6.1	1.9 ± 0.9	0.0 ± 0.0	40.4 ± 2.2 ^##^	817 ± 227	11.8 ± 3.7	12.9 ± 0.8
Female	Jimai22	2.7 ± 1.2	7.75 ± 0.48	144 ± 9	68.7 ± 4.9	2.1 ± 0.8	26.6 ± 4.3	2.7 ± 2.0	0.0 ± 0.0	41.3 ± 2.2	792 ± 161	12.5 ± 5.9	12.3 ± 0.5
	Control	2.8 ± 1.2	7.91 ± 0.54	149 ± 10	67.2 ± 8.0	2.0 ± 0.8	29.0 ± 8.6	1.9 ± 0.8	0.0 ± 0.0	43.7 ± 2.8	846 ± 109	12.2 ± 0.8	12.8 ± 0.5
	Gm	4.9 ± 1.6	8.40 ± 0.28	146 ± 5 *	64.0 ± 4.8	2.4 ± 0.6	31.2 ± 4.6	2.4 ± 0.7	0.0 ± 0.0	41.7 ± 1.2 **^##^	894 ± 90	10.9 ± 0.8	12.8 ± 2.0
Male	Jimai22	4.9 ± 1.4	8.68 ± 0.49	151 ± 7	60.6 ± 6.9	2.8 ± 0.7	34.1 ± 6.7	2.5 ± 0.8	0.0 ± 0.0	43.4 ± 1.5	881 ± 105	11.5 ± 1.1	13.0 ± 0.41
	Control	5.5 ± 1.3	8.36 ± 0.34	148 ± 6	63.3 ± 4.5	2.2 ± 0.5	32.1 ± 4.4	2.1 ± 0.8	0.0 ± 0.0	44.3 ± 1.8	863 ± 99	12.5 ± 1.6	12.9 ± 0.57

* represents *p* < 0.05 compared with Jimai22 group; ** represents *p* < 0.01 compared with Jimai22 group; ^##^ represents *p* < 0.01 compared with control group.

**Table 2 plants-11-01823-t002:** Serum chemistry parameters of male and female rats ($\bar{x}$ ± s, *n* = 20).

Sex	Group	Glu	BUN	Cr	CHOL	TG	ALT	AST	TP	ALB	ALB/GLO	K^+^	Na^+^	Cl^−^	Ca^2+^
		(mmol/L)	(mmol/L)	(μmol/L)	(mmol/L)	(mmol/L)	(U/L)	(U/L)	(g/L)	(g/L)		(mmol/L)	(mmol/L)	(mmol/L)	(mmol/L)
	Gm	6.03 ± 0.98 ^#^	4.12 ± 0.57 *^##^	64.9 ± 5.5	2.04 ± 0.48 **^##^	0.50 ± 0.16 ^a^	31 ± 12 ^#^	85 ± 18 ^##^	60.8 ± 2.6 ^#^	32.3 ± 1.5	1.13 ± 0.05 ^a^	4.43 ± 1.57	140.2 ± 1.4	103.1 ± 1.5	2.54 ± 0.09 ^##^
Female	Jimai22	6.33 ± 0.95	4.72 ± 0.76	67.4 ± 7.4	1.64 ± 0.35	0.57 ± 0.18 ^a^	29 ± 6	86 ± 18	61.6 ± 3.0	33.1 ± 1.6	1.19 ± 0.08 ^a^	4.00 ± 0.32	139.9 ± 1.7	103.3 ± 1.4	2.53 ± 0.07
	Control	5.21 ± 0.96	5.22 ± 1.03	68.2 ± 7.9	1.59 ± 0.35	0.52 ± 0.16	39 ± 8	113 ± 12	58.2 ± 3.6	31.3 ± 1.9	1.17 ± 0.07	4.53 ± 0.52	140.0 ± 2.6	103.4 ± 1.6	2.44 ± 0.08
	Gm	6.85 ± 0.86 ^##a^	4.89 ± 0.55	61.4 ± 6.3	1.88 ± 0.56 *	0.60 ± 0.24	34 ± 6 ^a^	85 ± 9 ^#a^	57.4 ± 2.7	28.7 ± 1.2	1.00 ± 0.06	4.39 ± 0.68	139.7 ± 0.8 ^##^	100.8 ± 1.5 ^##^	2.53 ± 0.08
Male	Jimai22	6.62 ± 0.93 ^a^	4.71 ± 0.57	60.7 ± 5.1	1.52 ± 0.35	0.74 ± 0.46	32 ± 5 ^a^	90 ± 13 ^a^	58.2 ± 2.5	29.2 ± 1.4	1.01 ± 0.09	4.39 ± 0.47	140.2 ± 1.0	101.2 ± 1.6	2.52 ± 0.07
	Control	5.85 ± 0.84	4.51 ± 0.59	58.0 ± 4.5	1.76 ± 0.46	0.83 ± 0.38	38 ± 7	94 ± 12	57.7 ± 2.3	28.5 ± 1.01	0.98 ± 0.05	4.37 ± 0.34	143.2 ± 1.1	103.0 ± 1.2	2.50 ± 0.04

* represents *p* < 0.05 compared with Jimai22 group; ** represents *p* < 0.01 compared with Jimai22 group; ^#^ represents *p* < 0.05 compared with control group; ^##^ represents *p* < 0.01 compared with control group; ^a^ represents this data was cited in our last paper (Tian et al., 2021) [15].

**Table 3 plants-11-01823-t003:** Organ weights and relative organ weights of females ($\bar{x}$ ± s, *n* = 20).

Organ	Gm		Jimai22		Control	
	Weights (g)	Relative Weights (%)	Weights (g)	Relative Weights (%)	Weights (g)	Relative Weights (%)
Body weight	354.2 ± 36.4		348.4 ± 33.1		302.0 ± 28.2	
Heart	1.02 ± 0.13 ^#^	0.288 ± 0.034	1.02 ± 0.11	0.293 ± 0.029	0.93 ± 0.11	0.310 ± 0.035
Liver	9.32 ± 1.14 **^##^	2.648 ± 0.378 **	8.29 ± 0.84	2.379 ± 0.099	7.91 ± 1.04	2.631 ± 0.370
Spleen	0.75 ± 0.10 ^#^	0.213 ± 0.022	0.71 ± 0.09	0.205 ± 0.020	0.66 ± 0.10	0.220 ± 0.034
Kidneys	2.11 ± 0.018 ^##^	0.598 ± 0.056	2.06 ± 0.23	0.592 ± 0.059	1.89 ± 0.15	0.630 ± 0.061
Brain	1.89 ± 0.25	0.538 ± 0.088 ^##^	1.89 ± 0.06	0.546 ± 0.056	1.87 ± 0.08	0.623 ± 0.059
Uterus	0.62 ± 0.21	0.178 ± 0.068	0.70 ± 0.29	0.203 ± 0.084	0.61 ± 0.17	0.202 ± 0.056
Ovaries	0.21 ± 0.03	0.061 ± 0.009 ^#^	0.21 ± 0.04	0.059 ± 0.009	0.22 ± 0.04	0.073 ± 0.017
Adrenals	0.103 ± 0.024 *	0.030 ± 0.007 *	0.085 ± 0.018	0.024 ± 0.005	0.094 ± 0.026	0.031 ± 0.009
Thymus	0.461 ± 0.078 ^##^	0.131 ± 0.023	0.424 ± 0.139	0.122 ± 0.037	0.353 ± 0.086	0.117 ± 0.029

* represents *p* < 0.05 compared with Jimai22 group; ** represents *p* < 0.01 compared with Jimai22 group; ^#^ represents *p* < 0.05 compared with control group; ^##^ represents *p* < 0.01 compared with control group.

**Table 4 plants-11-01823-t004:** Organ weights and relative organ weights of males ($\bar{x}$ ± s, *n* = 20).

Organ	Gm		Jimai22		Control	
	Weights (g)	Relative Weights (%)	Weights (g)	Relative Weights (%)	Weights (g)	Relative Weights (%)
Body weight	587.1 ± 44.1		585.7 ± 69.5		524.0 ± 43.6	
Heart	1.49 ± 0.14	0.254 ± 0.023 ^#^	1.52 ± 0.17	0.261 ± 0.027	1.44 ± 0.17	0.275 ± 0.030
Liver	15.08 ± 1.80	2.577 ± 0.327 ^a^	14.80 ± 2.64	2.518 ± 0.254 ^a^	14.32 ± 1.31	2.740 ± 0.218
Spleen	1.11 ± 0.19	0.189 ± 0.030	1.05 ± 0.17	0.179 ± 0.021	1.17 ± 0.55	0.224 ± 0.103
Kidneys	3.46 ± 0.28	0.592 ± 0.055 ^a^	3.38 ± 0.41	0.579 ± 0.048 ^a^	3.25 ± 0.58	0.625 ± 0.122
Brain	2.10 ± 0.08	0.359 ± 0.033 ^#^	2.09 ± 0.09	0.361 ± 0.046	2.05 ± 0.08	0.394 ± 0.039
Testis	4.03 ± 0.53	0.688 ± 0.092	3.93 ± 0.81	0.672 ± 0.109	4.06 ± 1.18	0.773 ± 0.199
Epididymides	1.67 ± 0.19	0.287 ± 0.040	1.61 ± 0.21	0.278 ± 0.051	1.72 ± 0.62	0.329 ± 0.118
Prostate	1.49 ± 0.25	0.253 ± 0.038	1.51 ± 0.26	0.262 ± 0.058	1.39 ± 0.25	0.265 ± 0.043
Adrenals	0.090 ± 0.021	0.015 ± 0.004	0.084 ± 0.020	0.014 ± 0.003	0.091 ± 0.025	0.018 ± 0.005
Thymus	0.594 ± 0.168	0.101 ± 0.028	0.571 ± 0.152	0.097 ± 0.023	0.503 ± 0.119	0.096 ± 0.024

^#^ represents *p* < 0.05 compared with control group; ^a^ represents this data was cited in our last paper (Tian et al., 2021) [15].

**Table 5 plants-11-01823-t005:** Histopathological results of three groups at week 13.

Histopathological Lesions	Groups					
	Gm Group		Jimai22 Group		Control Group	
Sex	♀	♂	♀	♂	♀	♂
Number	20	20	20	20	20	20
**Lung**						
Slight interstitial inflammation	0	0	1	0	0	0
**Stomach**						
Slight glandular dilatation	0	0	0	1	0	0
**Liver**						
Slight inflammatory cell infiltration	0	1	1	1	0	0
Slight vacuolation	1	0	0	1	0	0
Severe vacuolation	0	0	0	1	0	0
**Kidney**						
Slight interstitial inflammatory cell infiltration	0	1	1	0	0	1
Slight hyaline casts	2	2	2	1	1	1
Slight calcium deposition	2	0	1	0	1	0
**Testis**						
Severe atrophy	-	1	-	2	-	1
**Epididymis**						
Sperm loss	-	1	-	2	-	1

“-”: Not applicable.

## Data Availability

The data presented in this study are availability on request from the corresponding author. The data are not publicly available due to privacy.

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
