# Peer review of "Subchronic Toxicity of GmDREB3 Gene Modified Wheat in the Third Generation Wistar Rats"

_plants, 2022, doi:10.3390/plants11141823_

Round 1

Reviewer 1 Report

1.         This study was conducted to evaluate the subchronic toxicity of GmDREB3 gene modified wheat in the third generation rats. However, it is not known how many dosages (% or mg/kg) of transgenic wheat diet (Gm), parental wheat diet (Jimai22) were used in diet to fed Wistar rats? What information or references were used to determine the dosage of GM wheat? Please indicate the percentage or ppm of GM wheat and Jimai22 in the feed in this study.

2.         The homogeneity and stability of test articles (Gm) in AIN93 diet should be provided.

3.         This research was in compliance with OECD Guideline 408 for repeated dose 90-day toxicity study in rodents. Line 32: the OECD guideline 408 has been revised since 2018, thus it is recommended to cite the updated version. However, at least three dose levels and a concurrent control shall be used, except where a limit test is conducted (see paragraph 18).

4.         Material and methods: Urinalysis, Hematology, Clinical biochemistry and Histopathology. In comparison of the OECD guideline and present study, the urine appearance and volume are absent.

5.         Serum total T4, T3 and TSH should be measured on samples obtained from each animal in the main group and in satellite and/or recovery groups at study termination.

6.         It is recommended that significant difference symbols be marked in Figure 1, Figure 3 and Figure 5.

7.         Please add the semi-quantitative definition of histopathological examination in the materials and methods section. The incidence and semi-quantitative score system are recommended by Shackelford et al. (2002). The degree of lesions in each item is graded from one to five depending on severity: 1 = minimal (< 1%); 2 = slight (1-25%); 3 = moderate (26-50%); 4 = moderate/severe (51-75%); 5 = severe/high (76-100%).

8.         The results from the present study demonstrated that 13 weeks’ consumption of Gm wheat did not cause any adverse effects in the third generation rats when compared with the corresponding Jimai22 wheat. Please define the level of NOAEL (?? mg/kg/day) in rats and margin of safety for human (acceptable daily intake, ADI, ?? mg/kg/day) of Gm based on NOAEL in the 90-day study?

9.         The title of Figure 6 can change to be “Non-specific histopathological changes in rats at week 13”, or maybe not necessary to show these photos.

10.     The format of references needs to be rechecked, such as: delete underline….

Author Response

Dear Reviewer,
Thank you very much for your patient review of our manuscript (Plant-1775814) entitled “Subchronic toxicity of GmDREB3 gene modified wheat in the third generation Wistar rats”. These comments are very valuable and helpful for improving our paper. We have studied comments carefully and have made revisions. We hope these revisions can make our paper meet approval requirement. Revised portions are marked with revising format. The main corrections in the paper and the responses to comments are as follows: 

  1. This study was conducted to evaluate the subchronic toxicity of GmDREB3 gene modified wheat in the third generation rats. However, it is not known how many dosages (% or mg/kg) of transgenic wheat diet (Gm), parental wheat diet (Jimai22) were used in diet to fed Wistar rats? What information or references were used to determine the dosage of GM wheat? Please indicate the percentage or ppm of GM wheat and Jimai22 in the feed in this study.

Response: Thank you for your comment. Gm and Jimai22 wheat were both formulated into diets at a ratio of 69.55% according to AIN93 diet for rodent animals, and the information was added in the revised version. The energy density of these three diets were nearly identical (AIN93:1445.5 kJ/100g; Jimai22: 1550.6 kJ/100g and Gm 1504.9kJ/100g). The dosage of Gm wheat was determined by the contents of carbohydrate, fat and protein in AIN-93 diets(Reeves. et al.,1993).

  1. The homogeneity and stability of test articles (Gm) in AIN93 diet should be provided

Response: Thanks for your comment. It was very difficult to determine the homogeneity and stability of Gm wheat in diets according to a certain chemical content, but the target gene was detected by PCR in Gm diet of different batch diets and different part of the same batch diets to prove the homogeneity and stability ( Tian. et al., 2019) .

  1. This research was in compliance with OECD Guideline 408 for repeated dose 90-day toxicity study in rodents. Line 32: the OECD guideline 408 has been revised since 2018, thus it is recommended to cite the updated version. However, at least three dose levels and a concurrent control shall be used, except where a limit test is conducted (see paragraph 18).

Response: Thank you for your comment. This study was performed in 2016, therefore, we didn’t refer to OECD guideline 2018. During the trial design phase,we presumed the possibility of  toxicity of Gm wheat is very low, so we adopted the maximum Gm dose in animal diets to meet AIN-93 formula requirement. In addition, we also consulted some references for animal experiments for transgenic crops, in which basic control diet group, non-Gm group and Gm group also were set. We will strictly refer to OECD guideline 408 (2018 revised version ) in the future study for 90-day subchronic toxicity.

  1.  Material and methods: Urinalysis, Hematology, Clinical biochemistry and Histopathology. In comparison of the OECD guideline and present study, the urine appearance and volume are absent.

Response: Thank you for your suggestion. According to OECD requirements, the urine appearance and volume must be present. The urine appearance and volume, which were recorded in primary data, have been supplemented in revised manuscript.

  1. Serum total T4, T3 and TSH should be measured on samples obtained from each animal in the main group and in satellite and/or recovery groups at study termination.

Response: Thank you for your comment. Serum total T4, T3 and TSH in this study were not measured. We acknowledge it was really a limitation of the study.

  1.    It is recommended that significant difference symbols be marked in Figure 1, Figure 3 and Figure 5.

Response: Thank you for your suggestion. We have revised Figure 1, Figure 3 and Figure 5.

  1. Please add the semi-quantitative definition of histopathological examination in the materials and methods section. The incidence and semi-quantitative score system are recommended by Shackelford et al. (2002). The degree of lesions in each item is graded from one to five depending on severity: 1 = minimal (< 1%); 2 = slight (1-25%); 3 = moderate (26-50%); 4 = moderate/severe (51-75%); 5 = severe/high (76-100%).

Response: Thank you for your suggestion. We have added the semi-quantitative definition of histopathological examination in the materials and methods section.

  1. The results from the present study demonstrated that 13 weeks’ consumption of Gm wheat did not cause any adverse effects in the third generation rats when compared with the corresponding Jimai22 wheat. Please define the level of NOAEL (?? mg/kg/day) in rats and margin of safety for human (acceptable daily intake, ADI, ?? mg/kg/day) of Gm based on NOAEL in the 90-day study?

Response: Thank you very much. In this study, Gm wheat and Jimai22 both accounted for 69.55% in animal diets, and food consumption of rats in 90-day subchronic study is about 80g/kg/day, that is, the average consumption of the Gm wheat of rats was approximately 55.64 g/kg/day (calculated as 80g/kg BW multiply 0.6955), so the level of NOAEL in rats was 55.64 g/kg/day. Because the equivalent effect doses in rats and human is 5:1, so, margin of safety for humans of this Gm wheat is 11.13g/kg/day.

  1. The title of Figure 6 can change to be “Non-specific histopathological changes in rats at week 13”, or maybe not necessary to show these photos.

Response: Thank you for your suggestion. We have changed the title of Figure 6 in revised manuscript.

  1. The format of references needs to be rechecked, such as: delete underline….

Response: Thank you for your suggestion. We have revised the format of references.

Sincerely yours,

Jie Tian

Reviewer 2 Report

The manuscript is a well written, interesting feeding study on 3rd generation rats fed with GM wheat or a diet made from conventional wheat. In the following I offer some general comments for further improvement. My major comments are directed to better describing the diets used in the trial. I, however, refrain from an in-depth review of the statistical analysis of the results of the described experiment - suggesting that another reviewer offers advice on this part. Further a number of minor editorial changes need to be made, some which are pointed out in the following detailed comments.

Detailed comments

Suggested changes are indicated by page no, paragraph number and number of line in the indicated para.

Abstract

P1, L4 Consider changing “reserved” for “used in this study”.

Introduction

P1, Para1, L1 Delete superfluous blank ahead of comma.
Mind other numerous instances where such superfluous blanks are included in the current text. (P10, Para2, L7)

P1, Para2, L3-4: I would not subscribe to the conclusion drawn in this sentence in a general sense. On the contrary I believe that conventional breeding has been quite effective to address complex quality traits such as adaption to changing climate conditions, which are usually mediated by multiple genetic factors. The authors are however right that such breeding and selection approaches require substantial efforts and investments and are usually slower than biotechnology interventions. Pls. revise accordingly!

P1, Para2, L4-8: I don´t think that this sentence captures the conclusions of the referenced article well. Indeed, the article points out the complexities and challenges of developing “draught-tolerant” varieties, irrespective of the technologies applied during breeding. Consider the overall take of the cited paper: “After more than 20 years of research and investment only a few such products have reached the market. This is due to several technical and market constraints. The technical challenges include the difficulty in defining tractable single-gene trait development strategies, the logistics of moving traits from initial to commercial genetic backgrounds, and the disconnect between conditions in farmer's fields and controlled environments.” Pls. revise the sentence to match the results of the cited study!

P2, Para1, L2:  Delete “kind of”; revise the term “dilemma” (challenges connected to an assessment of the food safety…)

P2, Para2, L2: Insert/delete blank: (this13) / (in SPF).

Chapter 2 Results

P2, Para3, L2: Revise spelling to “weights”

P2, Para3, L3 and following test: Check for missing articles and revise accordingly: e.g. to “with the Jimai22 group”

P4, Para1, L2: Revise spelling from “showed” to “shown” – also ff such as P4, Para2, last line; P8, Para1

P4, Para2, L1: You use the terms “decreased” and “increased” a lot in the following descriptions of results: This is a fine point, but since the data only address one time point the correct expressions would rather be: “lower (or higher)” values (were recorded). Pls. correct throughout! (P7, Para1;

P8, Para1, L1: Delete “kinds of”; indicate which group the 3 animals belong to!

Chapter3, Discussion

P9, Para1, L2: Revise blanks, change “is “ for “are”

P10, Para1, L4: End sentence after ref(19). Start new sentence “DREBs are also involved---“.

P10, Para1, L6: Pls. revise to “…widely used to genetically engineer plants for resistance to abiotic factors”-“.

P10, Para1, L8: Pls. revise to: “The GM wheat used in this study …”

P10, Para2, L2: Pls. revise “former” to “preceeding”

P10, Para2, L8: Pls. revise expression “one generation animals” (maybe: animals from a single generation??)

P10, Para2, L8: Pls. revise to “that long-term and multi-generation assessments are necessary …”

P10, Para2, L10: Pls. revise to: “So far, no 90-day subchronic study has been performed to address potential toxicological effects in animals exposed to GM feed over three generations”

P10, Para2, last line: Revise “ancestor” (parental animals??)

P10, Para3, L3: Pls. revise “contrast” to “control”

P10, Para3, L6: Pls. delete “as” and revise to “a normal fluctuation of body weights”

P10, Para4, L7: Pls. revise to: “ …were considered to be of no toxicological significance since these values were similar to the results recorded for the control group or the Jimai22 group.” Start new sentence with “In addition…”

P10, Para5, L1: Pls revise “increases” see above P4, Para2, L1

P10, Para6, L4: Pls. revise to: “since the respective findings were not supported by the biochemistry or histopathological data generated in our study.”

P10, Para6, last line: Revise “anatomy” to “necropsy”

P10, Para7, L1: Revise to “the GM group, while the severity and frequency of these lesions are comparable between the GM and Jimai22 groups”. Start new sentence: “Furthermore the incidence of these findings are within the normal reference range of such data established in our laboratory.”

P10, Para7, last line: delete “cases” and revise to “were concluded to be of no significance related to toxicity.”

Chapter 4 Materials and methods

P11, Para1: Pls revise “given” to “provided”.
Indicate whether the used test materials were analytically checked for identity and homogeneity.
Indicate the percentage of the wheat ingredient in the GM/Jimai22diets.

Indicate whether the formulation of these diets were similar throughout feeding of the test generation and the previous 2 generations.

Author Response

Dear  reviewer,

Thank you very much for giving us some precious comments and suggestions for revising our manuscript (Plant-1775814) entitled “Subchronic toxicity of GmDREB3 gene modified wheat in the third generation Wistar rats”. These comments are very valuable and helpful for improving our paper. We have studied your comments carefully and have made revisions. We hope these revisions can make our paper meet approval requirement. Revised portions are marked with revising format.

Sincerely yours,

Jie Tian

Round 2

Reviewer 1 Report

No more question.